# Association between Citrullinated Histone H3 and White Matter Lesions Burden in Patients with Ischemic Stroke

**DOI:** 10.3390/brainsci13070991

**Published:** 2023-06-24

**Authors:** Xiaohao Zhang, Yunzi Li, Zhenqian Huang, Shuaiyu Chen, Yan E, Yingdong Zhang, Qingguang Wang, Tingting Li

**Affiliations:** 1Department of Neurology, Nanjing First Hospital, Nanjing Medical University, Nanjing 210002, China; zxh_neurology@126.com (X.Z.); 18263707919@163.com (S.C.); joe970708@163.com (Y.E.); 2Department of Neurology, Jinling Hospital, Medical School of Nanjing University, Nanjing 210002, China; lyz3791207@163.com (Y.L.); njhzq2135@163.com (Z.H.); 3Department of Neurology, Jiangyin Hospital Affiliated to Nantong University, Jiangyin 214400, China; 4Department of Neurology, The Second Affiliated Hospital of Nanjing University of Chinese Medicine, Nanjing 210017, China

**Keywords:** CitH3, neutrophil extracellular traps, ischemic stroke, white matter lesions

## Abstract

Introduction: Neutrophil extracellular traps play a role in the pathophysiology of stroke and are associated with severity and mortality. We aimed to investigate whether the citrullinated histone H3 (CitH3), a biomarker for neutrophil extracellular traps formation, is associated with the white matter lesion (WML) burden in ischemic stroke patients. Methods: Between September 2021 and April 2022, 322 patients were enrolled in this prospective observational cohort study. Serum CitH3 levels were measured after admission using an enzyme-linked immunosorbent assay. WMLs severity was graded according to the Fazekas scale and conceptually defined as mild (total Fazekas score 0–2) and severe (total Fazekas score 3–6). We used multivariable regression models to determine the relationship between CitH3 concentrations and the severity of WMLs burden. Results: One-hundred and forty-eight (46.0%) patients were diagnosed with severe WMLs burden after admission. Increased CitH3 levels (first quartile vs. fourth quartile of H3Cit, odds ratio, 3.311, 95% confidence interval, 1.336–8.027; *p* = 0.011) were independently associated with a greater WML burden in the fully adjusted multivariable model. Similar results were found when the H3Cit was analyzed as a continuous variable. Furthermore, the multiple-adjusted spline regression model showed a linear association between H3Cit levels and severe WMLs (P = 0.001 for linearity). Conclusions: In the present study, increased CitH3 levels were positively associated with extensive WMLs in ischemic stroke patients, indicating a role of neutrophil extracellular traps formation in the pathogenesis of WMLs.

## 1. Introduction

White matter lesions (WMLs) are white matter abnormalities which are often observed on brain magnetic resonance imaging (MRI) in ischemic stroke patients, particularly if they are older and have several vascular risk factors [1,2]. WMLs are related to various geriatric syndromes, such as cognitive impairment, depressive mood, and balance and gait speed disorder [3,4,5]. Furthermore, it is now well-established that these WMLs are associated with poor functional outcome, increased mortality, higher risk of hemorrhage after thrombolysis, and long-term recurrence after ischemic stroke [6,7,8,9,10,11]. To date, WMLs are generally considered to be due to small vessel disease; however, the underlying pathological mechanisms behind this association are still unclear. Accurate identification of novel risk factors to improve the prediction of WMLs is highly desirable to elucidate the WML etiology and rapidly optimize patient management.

Neutrophil extracellular traps (NETs) are stranded, decondensed DNA fibers decorated with granular enzymes that are released from activated neutrophils [12,13,14]. NETs have also been reported to promote tissue damage, increase thrombosis, and cause endothelial dysfunction [15,16,17]. A critical step in NETosis is the citrullinated histone H3 (CitH3), a process that is mediated by protein deiminase 4, and CitH3 has been proposed as a target biomarker reflecting NET formation [18]. Recently, data from the Vienna Cancer and Thrombosis Study (CATS) demonstrated that a 100 ng/ML increase in CitH3 levels was associated with a 13% relative increase in venous thromboembolism risk in cancer patients [19]. Furthermore, prior studies have shown that elevated levels of CitH3 at onset were independently associated with atrial fibrillation and all-cause mortality at 1 year follow-up [20]. However, the relationship between CitH3 levels and WMLs burden is still unproven.

The present study aimed to evaluate whether the levels of CitH3, the most specific marker of NETs, were associated with the severity of WMLs burden in patients with ischemic stroke patients.

## 2. Methods and Materials

### 2.1. Study Sample

This is a single-center, prospective, and observational study. Patients diagnosed with ischemic stroke were consecutively enrolled between September 2021 and April 2022. The exclusion criteria of this study were as follows: (1) age less than 18 years, (2) hospitalized within 14 days after symptoms onset, and (3) had a National Institute of Health Stroke Scale score (NIHSS) ≤ 8. We excluded patients who were unable to perform the magnetic resonance imaging examination. Furthermore, patients with a history of leukodystrophy, demyelinating disease, central nervous system infection, autoimmune disease, and hematological disease were also excluded from this study. The study obtained the approval of the ethics committee of the Jinling Hospital (2021DZGZR-YBB-115) in compliance with the Declaration of Helsinki. All subjects signed informed consent before entering the study.

### 2.2. Baseline Data Collection

Baseline variables including age, sex, blood pressure, medication history, admission NIHSS score, and pathogenic stroke subtype determined using the criteria of Trial of ORG 10172 in Acute Stroke Treatment (TOAST) [21], and laboratory data were recorded after admission. We also collected data on vascular risk factors including hypertension (blood pressure ≥140/90 mmHg on repeated measurements or prior use of antihypertensive medication), diabetes mellitus (fasting blood glucose level ≥ 7.0 mmol/L on repeated measurements or the use of medications to lower blood glucose), and hyperlipidemia (total cholesterol level ≥ 5.7 mmol/L, triglyceride level ≥ 1.7 mmol/L, low-density lipoprotein-cholesterol level ≥ 3.6 mg/dl, and/or having received treatment for dyslipidemia).

### 2.3. CitH3 Levels Assessment

Blood samples (5 mL) were drawn from the cubital vein within 24 h after admission and processed under standard laboratory procedure. Commercially available ELISA kits were used to quantify serum CitH3 (Cat#EPT-P-3097-48, Colorimetric). All samples were measured by a laboratory technician who was blinded to any clinical information of the patients.

### 2.4. Image Acquisition and Analysis

Brain magnetic resonance imaging was performed within 7 days of admission, including T1-weighted imaging, T2-weighted imaging, and fluid-attenuated inversion recovery (FLAIR). WMLs were defined as supratentorial white matter hyperintensity on FLAIR according to the STandards for ReportIng Vascular changes on nEuroimaging criteria [22]. Disagreements of imaging analysis were resolved through a consensus conference. Furthermore, WMLs burden was graded according to the Fazekas scale [23] on the basis of the visual assessment of FLAIR images both periventricular (0: absent, 1: caps or pencil lining, 2: smooth halo, and 3: irregular periventricular hyperintensity extending into deep white matter) and subcortical areas (0: absent, 1: punctate foci, 2: beginning confluence of foci and 3: large confluent areas). A total Fazekas score, ranging from 0 to 6, was calculated by summing the periventricular and subcortical WMLs scores [24]. According to previous studies, severe WMLs were defined as a total Fazekas score ≥ 3 [25,26,27].

### 2.5. Statistical Analysis

Continuous variables were presented as the mean ± standard deviations (SD) or median with interquartile range (IQR) and compared between groups using Student’s t-test or the Kruskal–Wallis H test. Data for categorical variables were demonstrated as percentages and compared between groups using the Fisher exact test or χ2 test. Binary logistic regression models were utilized to evaluate the relationship between CitH3 levels and WMLs degree. Model 1 was adjusted for age and sex. Model 2 was adjusted for model 1 adding variables with a significance level < 0.1 in univariate statistics. The results were shown as odds ratio (OR) and 95 % confidence intervals (CI). We also assessed the pattern and magnitude of the association of CitH3 with the WML burden using the restricted cubic splines with 3 knots (at 5th, 50th, and 95th percentiles) after adjustment for model 2 [28]. A 2-sided *p* value < 0.05 was considered statistically significant. Statistical analyses were performed with SPSS 25.0 (SPSS, Inc., Chicago, IL, USA) and R statistical software version 4.0 (R Foundation, Vienna, Austria).

## 3. Results

### 3.1. Demographic and Clinical Characteristics

In total, 322 patients who were diagnosed with ischemic stroke (185 men; mean age, 64.7 ± 11.6 years) were enrolled in this study. The median (IQR) CitH3 levels of this study population were 32.7 (13.2–67.1) ng/mL. Baseline characteristics of the included 322 patients as stratified by CitH3 quartiles were summarized in Table 1. According to the total Fazekas score, 144 (46.0%) patients were diagnosed with severe WMLs burden. Vascular risk factors were present as follows: hypertension in 233 patients (72.4%), diabetes in 127 (39.4%), ischemic heart disease in 54 (16.8%), and hyperlipidemia in 38 (11.8%). On the univariate analysis, hypertension, severe WMLs, large atherosclerotic stroke, homocysteine, and hypersensitive C-reactive protein levels differed significantly with increasing quartiles of CitH3 concentrations. Sex and the other vascular risk factors did not differ significantly among quartiles.

### 3.2. CitH3 Levels and WMLs Burden

On the univariate analysis, patients with severe WMLs were older than those without it (mean, 66.7 versus 63.0 years; *p* = 0.004). Baseline systolic blood pressure was higher in patients with severe WMLs than in patients without (mean, 145.0 versus 137.9 mmHg; *p* = 0.005). Large atherosclerotic stroke was more prevalent in patients with extensive WMLs than in patients without (54.7% versus 42.5%; *p* = 0.012). Furthermore, patients with severe WMLs had higher levels of homocysteine (mean, 15.1 versus 12.3 mmol/L; *p* = 0.001) and CitH3 (median, 45.2 versus 19.6 ng/mL; *p* = 0.001) (Table 2).

Table 3 shows the results of the comparison of regression analysis of the association between CitH3 levels and WML burden. After adjusting for age, sex, and variables with a *p* value < 0.1 in the univariate analysis (including systolic blood pressure, stroke subtypes, and homocysteine levels), increased CitH3 levels were associated with significantly a greater WML burden in ischemic stroke patients (Per 1-SD increase in CitH3, OR, 3.311, 95% CI, 1.336–8.027; *p* = 0.011). Similar results were found when the H3Cit was analyzed as a categorical variable. Furthermore, the multiple-adjusted spline regression model showed a linear association between H3Cit levels and severe WMLs (*p* = 0.001 for linearity; Figure 1).

## 4. Discussion

This prospective study examined the association of CitH3 levels with WML burden in a population sample of ischemic stroke. A higher level of CitH3 concentrations was associated with an increased risk of severe WMLs in a dose–response manner. These positive associations remained significant after controlling for age, sex, systolic blood pressure, stroke subtypes, and homocysteine levels. These findings indicate a pivotal role of neutrophil extracellular traps formation in the presence and progression of WMLs.

Our results showed a prevalence of 46.0% for severe WMLs, which was in parallel with previous research [29,30], while higher than the findings (36.4% for severe WMLs) by Suda and colleagues [31]. Additionally, previous studies have demonstrated that increasing age, hypertension, and baseline blood glucose were prominent risk factors for WMLs [29,30,31]. However, our study did not observe any associations of vascular risk factors with WMLs. We supposed that differences in the study population may be the main reason for this discrepancy.

CitH3 has been developed as a biomarker for NETs formation in different experimental models, as citrullination of histone H3 by peptidyl arginine deiminase 4 causes the chromatin decondensation and subsequent NETs formation [32]. Recently, several clinical and experimental studies have confirmed the detrimental effect of NETs in cerebrovascular disease [20,33,34]. In a prospective cross-sectional study enrolling 243 ischemic stroke patients, Vallés and colleagues found that higher levels of citH3 were independently associated with all-cause death at 12-month follow-up [19]. Additionally, Zhang et al. [34] showed that after thrombolysis, a significant elevated NETs marker was observed in no-improvement patients, while the changes in improvement patients were not significant. Additionally, targeting NETs protected mice from tMCAO-induced cerebral ischemia, possibly by mediating von Willebrand factor and plasminogen activator inhibitor-1 in the blood and thrombi. Our study extended the current knowledge about the detrimental effect of NETs in cerebrovascular disease as it showed a negative association between CitH3 levels and WML burden in patients with ischemic stroke. Future studies are recommended to detect the benefit of regulating NETs in ischemic stroke patients with WMLs.

Although the mechanisms for WMLs are yet to be elucidated, there are several possible pathophysiological pathways for the relationship between high CitH3 levels and WMLs that have been suggested. Firstly, endothelial dysfunction is likely to represent an important link between these two conditions. An abnormal increase of CitH3 can damage human umbilical vein endothelial cells and induce endothelial cell leakage [16]. Given the pivotal role of the endothelium in cerebral circulation, endothelial dysfunction may theoretically contribute to histopathological cerebral parenchymal alterations observed in WMLs [35,36]. Secondly, it may be the result of atherosclerotic vessels. As demonstrated in our study, patients with increased CitH3 concentrations showed a higher ratio of large atherosclerotic stroke. Previous studies have confirmed that diffuse hypoperfusion due to major vessel steno-occlusion could worsen WMLs progression [37]. Finally, it was widely accepted that thrombosis and inflammation form a loop of bidirectional regulation, which may contribute to ischemic damage in white matter [38,39]. Interestingly, in previous literature studying Type 2 diabetes mellitus patients, NETosis detectable in circulating blood is associated with inflammatory state and a prothrombotic state, especially hypofibrinolysis [40]. We therefore speculated circulating CitH3 may induce the presence and progression of WMLs via thrombo-inflammation.

This study should be interpreted with caution because of several limitations. Firstly, patients with severe neurological deficits (baseline NIHSS score >8) were excluded from this study, which might induce a selective bias and underestimate the real incidence of WMLs. Secondly, the WML assessment was performed using a visual grading system. Although also widely used, the visual rating may be less precise or consistent than a quantitative manner. Thirdly, the direct NETs quantification has not yet been standardized, which hampered its applicability in a large clinical study. We therefore did not directly quantify NETs in the circulation, but indirectly via biomarkers reflecting NETs formation. Additionally, our study only measured CitH3 levels at baseline and did not examine the dynamic changes of CitH3, which may have provided more valuable information about the mechanism underlying the relationship between CitH3 and WMLs burden in ischemic stroke patients. Thus, future longitudinal studies evaluating multiple time points are desirable for validating the role of CithH3 in WMLs.

## 5. Conclusions

In conclusion, the findings in our study demonstrated that increased CitH3 levels were associated with greater WMLs in patients with ischemic stroke. Our study therefore further establishes higher CitH3 levels as a strong risk factor for WML burden and implies that treatment of CitH3 could lead to less WML progression in the ischemic stroke population. Future studies are warranted to validate the association and to further determine the exact mechanism linking CitH3 and WMLs.

## Figures and Tables

**Figure 1 brainsci-13-00991-f001:**
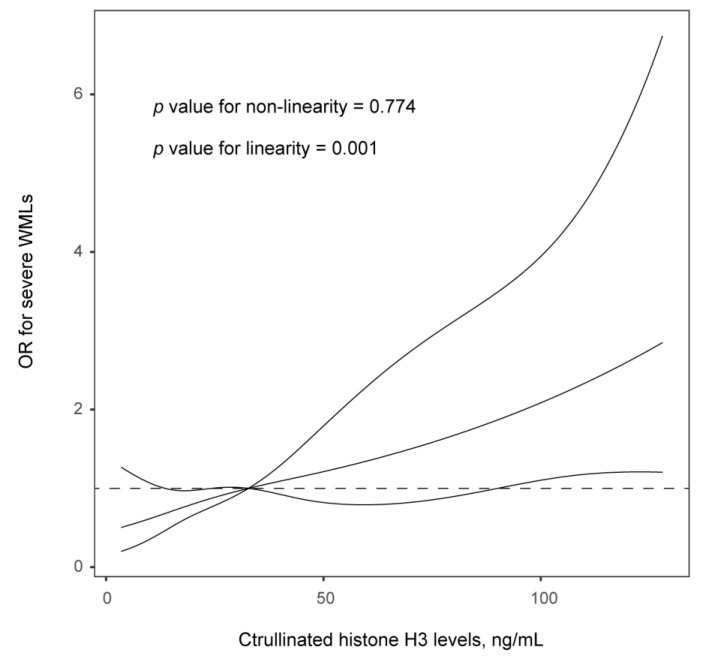
Association of serum citrullinated histone H3 (CitH3) with risk of severe WMLs in ischemic stroke patients. Odds ratios and 95% confidence intervals derived from restricted cubic spline regression, with knots placed at the 5th, 50th, and 95th percentiles of the distribution of CitH3 levels. The reference point for serum CitH3 is the midpoint (32.7 ng/mL) of the reference group from categorical analysis. Odds ratios were adjusted for age, sex, and variables with a *p* value < 0.1 in the univariate analysis.

**Table 1 brainsci-13-00991-t001:** Baseline characteristics of the studied patient population as stratified by CitH3 levels.

Variable	1st Quartile *n* = 80	2nd Quartile *n* = 81	3rd Quartile *n* = 80	4th Quartile *n* = 81	*p* Value
Demographic characteristics					
Age, years	63.9 ± 10.8	64.8 ± 11.5	64.4 ± 12.4	65.6 ± 11.8	0.811
Male, *n* (%)	39 (48.8)	46 (56.8)	48 (60.0)	52 (64.2)	0.239
Vascular risk factors, *n* (%)					
Hypertension	53 (66.3)	59 (72.8)	52 (65.0)	69 (85.2)	0.016
Diabetes mellitus	35 (43.8)	29 (35.8)	29 (36.3)	34 (42.0)	0.653
Hyperlipidemia	8 (10.0)	10 (12.3)	13 (16.3)	7 (8.6)	0.463
Coronary heart disease	12 (15.0)	17 (21.0)	11 (13.8)	14 (17.3)	0.626
Smoking	39 (48.8)	33 (40.7)	26 (32.5)	33 (40.7)	0.223
Clinical data					
Previous statin therapy, *n* (%)	20 (25.0)	21 (25.9)	27 (33.8)	32 (39.5)	0.149
Previous antiplatelet therapy, *n* (%)	18 (22.5)	27 (33.3)	26 (32.5)	30 (37.0)	0.228
Systolic blood pressure, mmHg	144.0 ± 23.8	137.4 ± 23.0	142.7 ± 25.2	140.5 ± 18.5	0.268
Diastolic blood pressure, mmHg	83.7 ± 11.8	79.9 ± 13.5	83.3 ± 15.1	81.5 ± 9.8	0.213
Baseline NIHSS, score	2.0 (0, 4.0)	3.0 (0, 4.0)	2.0 (0, 4.0)	2.0 (0, 5.0)	0.319
Severe white matter lesions	25 (31.3)	31 (38.3)	38 (47.5)	54 (66.7)	0.001
Stroke etiology, *n* (%)					0.007
Large artery atherosclerosis	29 (36.3)	38 (46.9)	40 (50.0)	48 (59.3)	0.034
Cardioembolic	7 (8.8)	13 (16.0)	14 (17.5)	12 (14.8)	0.407
Small vessel occlusion	34 (42.5)	16 (19.8)	19 (23.8)	15 (18.5)	0.001
Others	10 (12.5)	14 (17.3)	7 (8.8)	6 (7.4)	0.197
Laboratory data					
Total cholesterol, mmol/L	3.8 ± 0.9	3.9 ± 0.8	3.8 ± 0.9	3.7 ± 1.0	0.464
Triglyceride, mmol/L	1.4 (1.1, 2.1)	1.3 (1.0, 1.9)	1.4 (1.1, 2.2)	1.4 (1.1, 2.0)	0.468
Low density lipoprotein, mmol/L	2.1 (1.4, 3.0)	2.0 (1.8, 2.4)	2.1 (0.9, 2.5)	1.8 (1.4, 3.4)	0.243
High density lipoprotein, mmol/L	1.0 (0.9, 1.1)	1.0 (0.9, 1.2)	1.1 (0.9, 1.2)	0.9 (0.8, 1.1)	0.117
Homocysteine, mmol/L	11.7 ± 3.1	13.7 ± 6.1	13.4 ± 4.3	15.5 ± 6.5	0.002
Baseline blood glucose, mmol/L	7.4 ± 2.9	6.9 ± 2.6	6.7 ± 2.6	6.5 ± 2.4	0.185
Hs-CRP, mg/L	2.3 (1.1, 4.4)	5.7 (1.8, 13.2)	4.7 (2.2, 7.9)	13.1 (3.8, 21.0)	0.001

Abbreviations: CitH3, Citrullinated histone H3; Hs-CRP, hypersensitive C-reactive protein; NIHSS, National Institute of Health Stroke Scal. *p* indicated the statistical difference in the baseline data between groups.

**Table 2 brainsci-13-00991-t002:** Comparison of baseline data in patients with and without severe WMLs.

Variables	Severe WMLs		*p* Value
	Yes, *n* =148	No, *n* = 174	
Demographic characteristics			
Age, years	66.7 ± 12.5	63.0 ± 10.5	0.004
Male, *n* (%)	89 (60.1)	96 (55.2)	0.369
Vascular risk factors, *n* (%)			
Hypertension	113 (76.4)	120 (69.0)	0.140
Diabetes mellitus	61 (41.2)	66 (37.9)	0.548
Hyperlipidemia	17 (11.5)	21 (12.1)	0.872
Coronary heart disease	30 (20.3)	24 (13.8)	0.121
Smoking	61 (41.2)	70 (40.2)	0.857
Clinical data			
Previous statin therapy, *n* (%)	48 (32.4)	52 (29.9)	0.622
Previous antiplatelet therapy, *n* (%)	51 (34.5)	50 (28.7)	0.272
Systolic blood pressure, mmHg	145.0 ± 24.4	137.9 ± 20.9	0.005
Diastolic blood pressure, mmHg	83.4 ± 13.5	81.1 ± 11.9	0.106
Baseline NIHSS, score	2.0 (0, 4.0)	2.0 (0, 4.0)	0.476
Stroke etiology, *n* (%)			0.012
Large artery atherosclerosis	81 (54.7)	74 (42.5)	0.029
Cardioembolic	25 (16.9)	21 (12.1)	0.218
Small vessel occlusion	32 (21.6)	52 (29.9)	0.092
Others	10 (6.8)	27 (15.5)	0.014
Laboratory data			
Total cholesterol, mmol/L	3.8 ± 0.9	3.9 ± 1.0	0.170
Triglyceride, mmol/L	1.3 (1.0, 2.0)	1.4 (1.1, 2.2)	0.175
Low density lipoprotein, mmol/L	1.9 (1.5, 2.5)	2.1 (1.6, 2.6)	0.147
High density lipoprotein, mmol/L	1.0 (0.9, 1.2)	1.0 (0.9, 1.2)	0.278
Homocysteine, mmol/L	15.1 ± 6.0	12.3 ± 4.3	0.001
Baseline blood glucose, mmol/L	6.9 ± 2.3	6.8 ± 2.9	0.739
Hs-CRP, mg/L	5.2 (2.6, 11.5)	4.0 (1.3, 12.1)	0.124
CitH3, ng/mL	45.2 (17.8, 82.6)	19.6 (10.3, 46.5)	0.001

Abbreviations: CitH3, Citrullinated histone H3; Hs-CRP, hypersensitive C-reactive protein; NIHSS, National Institute of Health Stroke Scale; WMLs, white matter lesions. *p* indicated the statistical difference in the baseline data between groups. Informed consent was obtained from all subjects involved in the study. *p* indicated the statistical difference in the baseline data between groups.

**Table 3 brainsci-13-00991-t003:** Regression analysis of the association between CitH3 levels and WMLs burden.

Variables	Unadjusted OR (95%CI)	*p* Value	Adjusted OR (95%CI)	*p* Value
Per 1-SD increase in CitH3	1.75 (1.34–2.27)	0.001	1.67 (1.18–2.36)	0.004
CitH3 quartiles				
1st	Reference		Reference	
2nd	1.364 (0.711–2.616)	0.351	0.979 (0.380–2.526)	0.965
3rd	1.990 (1.044–3.794)	0.036	1.807 (0.721–4.532)	0.206
4th	4.400 (2.272–8.522)	0.001	3.311 (1.336–8.027)	0.011

Multivariate logistic regression analysis was adjusted for age, sex, and variables with a *p* value < 0.1 in the univariate analysis.

## Data Availability

The data that support the findings of this study are available on request from the corresponding author.

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
