# Peer review of "Association between Citrullinated Histone H3 and White Matter Lesions Burden in Patients with Ischemic Stroke"

_brainsci, 2023, doi:10.3390/brainsci13070991_

Round 1

Reviewer 1 Report

In the current study, the authors reported a positive correlation between Citrullinated histone H3 and extensive white matter lesions (WMLs) burden in ischemic stroke patients. This work is of significance, as any novel biomarkers that could improve the prediction of WMLs could facilitate rapid patient management.

Overall, the study is well designed with high quality data. The manuscript well written, providing sufficient background for the rationale. Just a few points:

1) Can the authors further elaborate the processing of the blood samples in session 2.3-CitH3 Levels Assessment? What tubes were the blood samples collected in, what volume of blood collected? Were serum or plasma used for the analysis?

2) For image analysis from 2.4: Image Acquisition and Analysis, can the authors provide more information on how the images were analysed, e.g. software etc

3) Figure 1: can the authors describe what the three lines represent in the figure legend

4) Is there any data on the effect of CitH3 levels on disease outlook e.g. fatality, recovery for the current study?

Author Response

In the current study, the authors reported a positive correlation between Citrullinated histone H3 and extensive white matter lesions (WMLs) burden in ischemic stroke patients. This work is of significance, as any novel biomarkers that could improve the prediction of WMLs could facilitate rapid patient management. Overall, the study is well designed with high quality data. The manuscript well written, providing sufficient background for the rationale. Just a few points:

1) Can the authors further elaborate the processing of the blood samples in session 2.3-CitH3 Levels Assessment? What tubes were the blood samples collected in, what volume of blood collected? Were serum or plasma used for the analysis?

Reply: Thanks for this comment. Blood samples (5 mL) were drawn from the cubital vein within 24 h after admission, and processed under standard laboratory procedure. Commercially-available ELISA kits were used to quantify the serum levels of CitH3. We now modified this description in the revised manuscript. (Line 89-91)

2) For image analysis from 2.4: Image Acquisition and Analysis, can the authors provide more information on how the images were analyzed, e.g. software etc.

Reply: Thanks for this advice. WMLs burden was graded according to the Fazekas scale on the basis of the visual assessment of FLAIR images both periventricular (0: absent, 1: caps or pencil lining, 2: smooth halo, 3: irregular periventricular hyperintensity extending into deep white matter) and subcortical areas (0: absent, 1: punctate foci, 2: beginning con-fluence of foci 3: large confluent areas).

3) Figure 1: can the authors describe what the three lines represent in the figure legend

Reply: The three lines in the figure legend indicated the OR and 95% CI.

4) Is there any data on the effect of CitH3 levels on disease outlook e.g. fatality, recovery for the current study?

Reply: Thank you for your comment. Recently, data from the Vienna Cancer and Thrombosis Study (CATS) demonstrated that a 100 ng/ML increase in CitH3 levels was associated with a 13% relative increase in venous thromboembolism risk in cancer patients (J Thromb Haemost. 2018;16(3):508-18). Furthermore, prior studies have shown that elevated levels of CitH3 at onset were independently associated with atrial fibrillation and all-cause mortality at 1-year follow-up (Thromb Haemost. 2017; 117(10):1919-29).

Reviewer 2 Report

Please clarify 

1)Methods and Materials: exclusion criteria??? of this study were as follows: (1) age older than 18 years; 

2) Image Acquisition and Analysis: Brain MRI was performed within 7 days of admission why 7 days?

3)Results/Demographic and Clinical Characteristics . On the 

univariate analysis, hypertension, severe WMLs, large atherosclerotic stroke, homocysteine, and hypersensitive C-reactive protein levels differed significantly with increasing 

quartiles of CitH3 concentrations, can you discuss de meaning of this results?

4) Please on table 1 and table 2 correct de position of p value in line stroke etiology 

Please clarify 

1)Methods and Materials: exclusion criteria??? of this study were as follows: (1) age older than 18 years; 

2) Image Acquisition and Analysis: Brain MRI was performed within 7 days of admission why 7 days?

3)Results/Demographic and Clinical Characteristics . On the 

univariate analysis, hypertension, severe WMLs, large atherosclerotic stroke, homocysteine, and hypersensitive C-reactive protein levels differed significantly with increasing 

quartiles of CitH3 concentrations, can you discuss de meaning of this results?

4) Please on table 1 and table 2 correct de position of p value in line stroke etiology 

'

Author Response

1) Methods and Materials: exclusion criteria??? of this study were as follows: (1) age older than 18 years; 

Reply: Thanks for your careful readings and comment. We are so sorry for this mistake. Patients with ages less than 18 years were excluded from this study. We now modified this description in the revised paper.

2) Image Acquisition and Analysis: Brain MRI was performed within 7 days of admission why 7 days?

Reply: Thanks for this comment. Patients with cerebrovascular diseases usually completed the MRI examination within 7 days in our hospital.

3)Results/Demographic and Clinical Characteristics. On the univariate analysis, hypertension, severe WMLs, large atherosclerotic stroke, homocysteine, and hypersensitive C-reactive protein levels differed significantly with increasing quartiles of CitH3 concentrations, can you discuss the meaning of this results?

Reply: Thanks for this comment. Previous studies have reported that an abnormal increase of CitH3 can damage human umbilical vein endothelial cells and induce endothelial cell leakage (Front Immunol. 2020; 10:2957). Interestingly, in previous literature studying Type 2 diabetes mellitus patients, NETo-sis detectable in circulating blood is associated with inflammatory state and a pro-thrombotic state, especially hypofibrinolysis (Cardiovasc Diabetol. 2019; 18:49). According to the above studies, we therefore speculated circulating CitH3 may be associated with the WMLs, large atherosclerotic stroke and inflammatory state.

4) Please on table 1 and table 2 correct deposition of p value in line stroke etiology 

Reply: Thanks for this helpful advice. We now correct the position of p value in line stroke etiology.

Reviewer 3 Report

Please add new references.

Introduction need to be modified to develop proper connection 

Minor correction

Author Response

Please add new references. Introduction need to be modified to develop proper connection 

Reply: Thanks for this comment. Several new references were added in the Introduction in the revised paper.